# SARS-CoV-2 Vaccination in Solid-Organ Transplant Recipients

**DOI:** 10.3390/vaccines10091430

**Published:** 2022-08-30

**Authors:** Maddalena Peghin, Elena Graziano, Paolo Antonio Grossi

**Affiliations:** Infectious and Tropical Diseases Unit, Department of Medicine and Surgery, University of Insubria-ASST-Sette Laghi, 21100 Varese, Italy

**Keywords:** SARS-CoV-2, transplant, COVID-19, vaccine, immunity

## Abstract

The coronavirus disease 2019 (COVID-19) pandemic has posed significant global challenges for solid organ transplant (SOT) recipients. Mortality rates of COVID-19 in this patient population remain high, despite new available therapeutic options and Severe Acute Respiratory Syndrome Coronavirus 2 (SARS-CoV-2) vaccination. Priority access to SARS-CoV-2 vaccination for waitlisted candidates and for SOT patients and their family members is recommended since the advantage from vaccination reduces the risk of COVID-19-related complications. However, immunogenicity and efficacy of COVID-19 vaccines are lower in waitlisted candidates and SOT recipients than in the general population. Routine systematic assessment of humoral and cellular immune responses after SARS-CoV-2 vaccination is controversial, although highly recommended for investigation and improvement of knowledge. SOT recipients should continue to adhere to preventive protective measures despite vaccination and may undergo passive antibody prophylaxis. This article seeks to provide an update on SARS-CoV-2 vaccination and preventive measures in SOT recipients based on existing literature and international guidelines.

## 1. Introduction

The coronavirus disease 2019 (COVID-19) pandemic has significantly changed solid organ transplantation (SOT) landscape worldwide [1]. At the beginning of the COVID-19 outbreak a considerable reduction in transplant activity and significant morbidity and mortality have represented the very first consequences of the pandemic on the transplant community [2]. The rapid development of vaccines against the SARS-CoV-2 has led transplant programs to return to pre-pandemic levels and to decrease COVID-19 morbidity and mortality in the SOT population [3,4]. SARS-CoV-2 vaccines have proven to be safe in SOT recipients but with decreased immunogenicity and efficacy compared with the general population. Moreover, the ongoing emergence of novel variants of concern (VOCs) remains a challenge, with reduced protection against infection, though SARS-CoV-2 vaccines seem to continue preventing severe illness resulting in hospitalization, intensive care admission, and death [5].

The aim of this article is to review the existing literature on SARS-CoV-2 vaccination and other preventive measures in SOT recipients.

## 2. The Impact of SARS-CoV-2 in SOT Recipients

The real impact of COVID-19 on SOT recipients has been debated since the very early reports. The influence of different factors has contributed to heterogenous results depending on the pandemic evolution through different waves, baseline characteristic of SOT recipients (age, ethnicity, comorbidities, type of organ transplanted and related net state of immunosuppression), available treatment options (use of corticosteroids, monoclonal antibodies and antiviral drugs) and SARS-CoV-2 vaccination [4].

Early reports and meta-analysis found variable rates in hospitalization (range 31–89%) intensive care unit (ICU) admission (range 11–33%) and mortality (range 5–27%) in SOT recipients depending on comorbidities and type of transplant [6,7,8].

A recent meta-analysis of 31 retrospective observational studies did not find a difference in 30-day mortality rates in SOT compared to the general population (OR 1.13, 95% CI, 0.94–1.35) but a higher risk of ICU admission (OR 1.56, 95% CI, 1.03–2.63) and of acute kidney injury (OR 2.50, 95% CI, 1.81–3.45) were reported [9]. In addition, even when no excess mortality was found, SOT recipients were found to be at a higher risk for respiratory failure and superinfections [6,10]. Prospective multicenter studies are needed to confirm this result and to stratify the risk according to the type of organ transplanted and evaluating the important role of the high burden of nonmodifiable risk factors such as comorbidities [7,8,10].

Of note, the available literature mainly comes from previous waves and VOC, and the impact of Omicron variant on SOT is still to be assessed. Moreover, the long-term impact on survival and on graft function of COVID-19 and of the reduction in immunosuppression needs to be more deeply understood [11,12], although some preliminary studies found no association with rejection and donor specific antibodies production in kidney transplant recipients after COVID-19 infection [13].

## 3. Available Vaccines and Recommendations

Numerous SARS-CoV-2 vaccines are available worldwide and several vaccine candidates are in development and some of them updated against new VOC [14]. Overall, the mRNA vaccines, including BNT162b2 (Pfizer-BioNTech COVID-19 vaccine) and/or mRNA-1273 (Moderna COVID-19 vaccine) have displayed higher protection than other types of vaccine against severe illness and mortality and against new VOC with an optimal safety profile [15]. Moreover, mRNA vaccines were approved before other vaccine types and majority of existing studies focused on COVID-19 vaccination in SOT recipients are based on the use of mRNA vaccines, rather than adenoviral vector vaccines (ChAdOx1 nCoV-19 vaccine OxfordAstraZeneca or Johnson & Johnson/Janssen) and inactivated whole virus vaccines (Sinovac-CoronaVac and Sinopharm) and protein-base vaccines (Nuvaxovid) [16]. Therefore, SARS-CoV-2 mRNA vaccines should be recommended for SOT and waitlisted patients. As an alternative, if mRNA vaccines are not available or suitable because of contraindications, adenoviral vector vaccines and inactivated whole virus vaccines should be recommended rather than renunciation of COVID-19 vaccination

## 4. Surrogate Measures of SARS-CoV-2 Vaccine Immunogenicity

Systematic assessment of immunogenicity after SARS-CoV-2 vaccination is controversial because, to date, there are no correlates of immune response and protection cut-offs against SARS-CoV-2 infection in the general and SOT populations [17,18]. The response to vaccination can be measured through different methods and assays according to the part of the adaptive immune system elicited by the vaccine, mainly including humoral and cellular immunity. It is of note that while vaccine-induced humoral immunity against SARS-CoV-2 is compromised by emergent VOC, cellular immunity against SARS-CoV-2 has shown to be robust against VOC [19].

The humoral response with the production of specific antibodies by B cells is detected with serologic techniques. The serological marker of SARS-CoV-2 vaccination is the presence of IgG against the receptor-binding domain (RBD) of the anti-SARS-CoV-2 recombinant Spike protein without the presence of IgG directed against the Anti-Nucleoside protein, produced during natural infection [20,21].

The presence of neutralizing antibodies, namely antibodies capable of binding to the free virus thus preventing the virus to infect cells, is considered of foremost importance in protection from viral infections. Although the neutralizing antibody level has more recently been established as a reliable predictor of protection against symptomatic COVID-19, the measures used in many studies varied as different techniques and different thresholds were identified [15].

The cellular immunity (driven by Spike-specific CD4+ T cells, Spike-specific CD8+ T cells, and Spike- and RBD-specific memory B cells) play a significant role in COVID-19 vaccine response but is still not completely understood, mainly because of the laborious diagnostic methods and diversity of cellular immunity [15]. Among assays to detect cellular response, simple laboratory tests including interferon-γ release assays (IGRAs) measure T lymphocytes interferon-γ production after stimulation by SARS-CoV-2 viral antigens and have shown potential in assessing the degree and durability of T-cell responses for better understanding of the protection offered by COVID-19 vaccination especially in susceptible populations [22].

## 5. Immunogenicity, Clinical Effectiveness and Safety of SARS-CoV-2 Vaccination in SOT

Pandemic and related vaccination efforts have been evolving, but SOT recipients were initially not included from the SARS-CoV-2 vaccine phase III trials, and data on clinical and immunological vaccine effectiveness in SOT recipients are deficient and most data come from retrospective and prospective real-world studies with variable immune response [23,24]. Moreover, there is a marked heterogeneity among series related type of SOT included in the studies (mostly kidney SOT), timing of booster vaccination doses, duration of follow-up and modality of assessment of immunogenicity and effectiveness. Lastly a significant limitation is that most available studies have been performed before the Omicron surge and that the risk of breakthrough infection is higher with certain variants, such as Delta and Omicron.

Based on research data and real-life experience, evolving vaccination strategies have been adapted to SOT population. Suboptimal immunogenicity and clinical effectiveness of standard two-dose vaccination strategy has led to administration of a third, and then a fourth vaccine dose to SOT recipients in many countries worldwide [25]. Humoral and cellular immune response, breakthrough infections and safety after one up to five SARS-CoV-2 vaccine doses are discussed in turn.

### 5.1. One-Two Doses of SARS-CoV-2 Vaccine

Rates of seroconversion after one and two doses of mRNA vaccines in SOT are lower than in the general population and variable ranging from 0% to 17% (vs. 26.9% of non-SOT controls) after the first dose and 21 to 64% after the second dose (vs. 100.0% of non-SOT controls) [23,24]. In addition, humoral immunity has shown to wane over time in the general population [26] and SOT recipients have shown to reach lower concentrations of antibodies and neutralizing activity at six months compared to general population [27,28].

Cardiothoracic transplant recipients seem to have the lowest rate of IgG production after two doses of mRNA vaccine (range 0–12% after the first dose and range 18–51% after the second dose), with higher rates of non-responders among lung recipients [29,30,31,32,33]. In heart recipients after two doses of ChAdOx1 nCoV-19 vaccine 24% seroconverted after one dose, increasing only to 34.8% after the second dose [34].

In studies comparing the serological responsiveness of the two mRNA vaccines, mRNA1273 vaccine outperformed the BNT162b2 both after a single dose (69% vs. 31% of seroconversion rate, *p* = 0.003) [35] and after the second dose with a higher rate of seroconversion (47% vs. 36%, *p* = 0.025) and a higher antibody titer [36,37].

It is of note that in studies including patients with a previous history of COVID-19 infections, seroconversion rates were sensibly higher even among unvaccinated patients, from 85% to 100% [30,34].

As for antibody production, data from a real-world study on cellular immunity response had variable results, confirming that also cellular immune response in SOT recipients is lower than in the general population and may be different depending on the type of mRNA vaccine. In a controlled cohort (1:1) of 200 SOT recipients, 13.1% of SOT population vs. 59.4% of healthy controls (*p* < 0.001) had a positive interferon-γ release assay 6 months after the second dose of BNT162b2 vaccine [28].

Two doses of mRNA-1273 vaccine in 58 liver and 46 heart recipients elicited 79% of ELISpot positivity 4 weeks after the second dose [37]. Interestingly, cellular response may be present in patients without detectable antibodies. In a cohort of lung transplant recipients with no seroconversion after two doses of mRNA BNT162b2 vaccine cellular response was detected [30].

In vaccinated SOT breakthrough infections occurred in 0.83–1.8% with a mortality rate ranging from 0 to 9.3% [38,39,40,41,42,43]. Although the overall incidence of breakthrough infections among vaccinated transplant patients is low, SOT vaccinated recipients have lower clinical effectiveness compared with immunocompetent individuals (82-fold higher risk of breakthrough infection and 485-fold higher risks of breakthrough infection with associated hospitalization and death) [40].

Overall SOT vaccinated (with two doses of mRNA vaccine or one Ad26.Cov2.S dose) compared to unvaccinated SOT, the vaccine has shown to decrease COVID-19 incidence rate, risk of symptomatic COVID-19, critical disease and mortality [25,30,44,45,46,47]. In contrast other studies reported that vaccination with one or two doses of m-RNA vaccine did not reduce risk of disease severity and death after SARS-CoV-2 infection in SOT recipients [48,49,50].

Vaccination in SOT shows the same high safety profile as in immunocompetent patients, with mild to moderate local or systemic side effects (pain at the injection site, fatigue, low-grade fever, headache) [23,31,37,51]. No graft dysfunction nor increase in HLA antibodies were observed after two doses of mRNA-1273 vaccine nor BNT162b2 vaccine [23,37].

### 5.2. Three Doses of SARS-CoV-2 Vaccine

Three doses of the COVID-19 messenger RNA vaccine is a minimum standard baseline recommended primary program for waitlisted and SOT recipients [52,53]. There is good evidence that seroconversion increased with a three-dose schedule compared to the standard two-dose. Overall, the effectiveness of the third dose of the vaccines has been observed to be about 90% for the infection rate in the immunocompetent population (in the pre-Omicron era) [54]. SOT patients have a less robust serologic response after booster vaccination but they still attain a significant benefit from the third dose, since the antibody titers increased after the third dose of about 27.4–50% of patients who had negative antibody titers and in 81.3–100% of patients who had low-positive antibody titers [55,56,57,58].

In addition, a third dose of mRNA-1273 vaccine has been observed to elicit neutralizing antibodies and specific CD4+ T cells count, including in primary no responders [59,60,61].

In regards to breakthrough infections, a recent Canadian study on SOT recipients, observed increasing effectiveness against infection (31%, 46% and 72%) and clinically important outcomes, such as hospitalization or death (38%, 54%, 67%) for one, two, and three doses, respectively. Overall, a third dose notably improved vaccine effectiveness against infection (72%) and clinically important outcomes (67%) [62]. Overall a third vaccination only reduces risk of SARS-CoV2 infection marginally, but SOT recipients vaccinated 3 times have reduced mortality (IRR: 0.22; 0.16–0.35) [53].

As regard as safety evaluation, local and systemic events have only a minor increase after the third dose of mRNA-1273 than after the dose of placebo but no grade 3 or 4 events and no cases of acute rejection have been described [57,59].

### 5.3. Four Doses of SARS-CoV-2 Vaccine

In most case series of SOT recipients receiving four doses of, SARS-CoV-2 vaccinated patients have been categorized as non-responders, low titer responders, or high titer responders after three doses of COVID vaccine based on antibody assay.

The fourth dose of a SARS-CoV-2 vaccine has been associated with a slightly improved humoral response among SOT patients with negative or low-positive humoral response. On the basis of available studies, a fourth mRNA vaccine dose have proven to improve humoral response only in a marginal number of previously seronegative SOT recipients (8–50%) and to boost antibody concentrations to potentially neutralizing levels against ancestral SARS-CoV-2 but not against Omicron VOC in a relevant part of previous low-responders (50–100%) [57,63,64,65,66,67].

There are limited data on outcome of breakthrough infections with Omicron VOC in SOT recipients who received four doses of vaccine. A recent longitudinal cohort on SOT with >3×-vaccines found that 10% of vaccinated SOT recipients developed COVID-19 during the Omicron wave. Interestingly seronegativity alone was not associated with breakthrough COVID-19, but incremental increase in antibody titers was associated with decreasing likelihood of reporting breakthrough infection [68].

A fourth mRNA vaccine dose in SOT has proven to be safe and well tolerated (no vaccine related serious adverse events and no rejection) and the benefit from vaccination may outweigh risk for most patients [57,63,64,65,66,67].

### 5.4. Five Doses of SARS-CoV-2 Vaccine

Currently, there are limited data evaluating the role of a fifth dose of SARS-CoV-2 vaccine [68]. In a German cohort of 1478 kidney SOT who received SARS-CoV-2 vaccination, authors observed that serological response was 19.5% after 1203 basic immunizations, and increased to 29.4%, 55.6%, and 57.5% in response to 603 third, 250 fourth, and 40 fifth vaccinations, resulting in a cumulative response rate of 88.7%. They found that repeated COVID-19 vaccination (five doses) was safe for the non-responding kidney SOT and induced sufficient serological response in most patients, except for belatacept-treated patients [69].

## 6. Risk Factors for Vaccine Unresponsiveness in SOT

In a recent meta-analysis of 29 studies including 11,713 SOT recipients’ factors associated with a negative humoral response after two doses of mRNA vaccines are deceased donor status, antimetabolites, calcineurin inhibitors, rituximab, anti-thymocyte globulin, while male sex and mTOR inhibitors were found to be associated with a better humoral response [70]. Other risk factor associated with mRNA vaccines unresponsiveness are older age, worse graft function, short time to transplantation (1 to 2 years) and betalacept use [28,31,35,37,51,71].

Additional risk factors for lack of seroconversion after two doses of BNT162b2 vaccine included diabetes, SOT other than liver transplant recipients (especially kidney and lung transplantation) corticosteroids combination of immunosuppressive drugs and de novo non-skin cancer comorbidity [23,28,51]. A reduced IgG production after two doses of mRNA-1273 vaccine was found to be also hypogammaglobulinemia [37]. A low seroconversion rate after two doses of ChAdOx1 nCoV-19 vaccine was seen in SOT recipients with chronic kidney disease and mycophenolate use [34].

## 7. Immunization before and after Transplant

The best approach to SARS-CoV-2 vaccination before and after transplant is not well known.

Waitlisted patients and SOT are at an increased risk of being infected with Severe Acute Respiratory Syndrome Coronavirus 2 (SARS-CoV-2), and to develop severe disease due to baseline comorbidities, impaired immune response, and contacts with the healthcare system [72].

Transplant waitlisted candidates should try to be vaccinated completely for COVID-19 at least 2 weeks prior to transplantation, and the simultaneous administration of other vaccines is considered to be safe. Patients with chronic liver disease and end-stage renal disease have shown a poor humoral and cellular immune response to SARS-CoV-2 vaccination compared with the general population [70,71,72,73,74]. However, vaccination of waitlisted candidates should be strongly encouraged for several reasons. First, transplant candidates and recipients are at increased risk of complications and death compared with the general population. Second, waitlisted patients who have been vaccinated exhibit a more robust response to vaccines, compared with the poor response in naive recipients vaccinated after transplantation. Third, vaccinating candidates might allow the use of organs from SARS-CoV-2-positive donors in recipients with prior SARS-CoV-2 vaccination. Fourth, vaccinated transplant candidates reduce the risk of COVID-19 transmission in the hospital setting to health care workers and other patients. Patient removal from the transplant list for refusing COVID-19 vaccination and vaccine mandates is a timely and controversial ethical topic [1,75].

The ideal timing of vaccination after transplantation is not known, and the probability of getting COVID-19 should be counterbalanced against the likelihood of developing an effective immune response after vaccination. For unvaccinated and not fully vaccinated patients, it is recommended to delay vaccination for at least one month from the time of transplantation, with a longer delay of at least three months in the case of the use of a T cell or B cell–depleting agent [76].

## 8. Modifications in Vaccination Strategies

To improve humoral responses to vaccination in SOT recipients several strategies other than booster doses have been attempted. Heterologous vaccination (i.e., vaccination with doses of different vaccines) has proven to induce at least as good a humoral response, if not a better response, to exclusive mRNA vaccination [77,78]. Otherwise, other strategies such as immunosuppression modulation to achieve immunity, timing of repeated doses, increased dosage administration are being evaluated and may be promising to improve poor immunological and clinical response [79].

## 9. Continued Use of Protective Measures and Potential Pre-Exposure Prophylaxis

Waitlisted patients and SOT recipients should continue to be recommended to maintain personal measures to minimise exposure to SARS-CoV-2 even after vaccination (hand hygiene, face masks, social distancing and vaccinating household members) and alternative strategies, including pre-exposure prophylaxis [14].

Passive antibody prophylaxis with tixagevimab plus cilgavimab is currently applicable for SOT patients older than 12 years (weighing ≥ 40 Kg) with no known current or recent COVID-19 exposure, who are receiving immunosuppressive treatment and that may have an inadequate immune response to vaccination or who cannot be vaccinated because of severe adverse reaction to COVID-19 vaccines [80]. A recent study confirmed that tixagevimab/cilgavimab use is safe and associated with a lower risk of breakthrough SARS-CoV-2 infection in vaccinated SOT during the Omicron wave [81]. Of note, that passive immunization should not be considered a substitute for vaccination and that failure may be associated with the development of SARS-CoV-2 VOC with reduced susceptibility to tixagevimab and cilgavimab [81].

## 10. Conclusions

Solid organ transplantation practice in the setting of the ongoing pandemic is an ever-evolving landscape, and it is vital for the transplant community to adapt to the changing circumstances. SOT recipients are at risk for developing COVID-19-related complications and for mortality, and they should be prioritized for SARS-CoV-2 vaccination. Vaccine effectiveness against SARS-CoV-2 infection is insufficient, but the benefit from vaccination and safety balances the risk.

There are several unmet needs that need yet to be addressed and based on research data and real-life experience; evolving vaccination strategies should be adapted to the SOT population.

## Data Availability

No new data were created or analyzed in this study. Data sharing is not applicable to this article.

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
