# Peer review of "SARS-CoV-2 Vaccination in Solid-Organ Transplant Recipients"

_vaccines, 2022, doi:10.3390/vaccines10091430_

Round 1
Reviewer 1 Report
I'd like to thank you for asking me to review this interesting paper about solid organ transplantation practice in the setting of the ongoing pandemic is an ever evolving landscape. SOT recipients are at risk for developing COVID-19-related complications and for mortality and they are at precedence for SARS-CoV-2 vaccination. Vaccine effectiveness against SARS-CoV-2 infection is insufficient but the benefit from vaccination and safety balances risk.
I suggest the authors to read these published papers to deep discuss your results:
-Coppeta L, Ferrari C, Mazza A, Trabucco Aurilio M, Rizza S. Factors Associated with Pre-Vaccination SARS-CoV-2 Infection Risk among Hospital Nurses Facing COVID-19 Outbreak. Int J Environ Res Public Health. 2021 Dec 10;18(24):13053. doi: 10.3390/ijerph182413053.
-Galanis P., Vraka I., Fragkou D., Bilali A., Kaitelidou D. Seroprevalence of SARS-CoV-2 antibodies and associated factors in healthcare workers: A systematic review and meta-analysis. J. Hosp. Infect. 2021;108:120–134. doi: 10.1016/j.jhin.2020.11.008.
-Coppeta L, Ferrari C, Somma G, Mazza A, D'Ancona U, Marcuccilli F, Grelli S, Aurilio MT, Pietroiusti A, Magrini A, Rizza S. Reduced Titers of Circulating Anti-SARS-CoV-2 Antibodies and Risk of COVID-19 Infection in Healthcare Workers during the Nine Months after Immunization with the BNT162b2 mRNA Vaccine. Vaccines (Basel). 2022 Jan 18;10(2):141. doi: 10.3390/vaccines10020141.
Author Response
Dear reviewer, we really appreciated your feedback and we have addressed all suggestions as required.
I'd like to thank you for asking me to review this interesting paper about solid organ transplantation practice in the setting of the ongoing pandemic is an ever evolving landscape. SOT recipients are at risk for developing COVID-19-related complications and for mortality and they are at precedence for SARS-CoV-2 vaccination. Vaccine effectiveness against SARS-CoV-2 infection is insufficient but the benefit from vaccination and safety balances risk.
I suggest the authors to read these published papers to deep discuss your results:
-Coppeta L, Ferrari C, Mazza A, Trabucco Aurilio M, Rizza S. Factors Associated with Pre-Vaccination SARS-CoV-2 Infection Risk among Hospital Nurses Facing COVID-19 Outbreak. Int J Environ Res Public Health. 2021 Dec 10;18(24):13053. doi: 10.3390/ijerph182413053.
-Galanis P., Vraka I., Fragkou D., Bilali A., Kaitelidou D. Seroprevalence of SARS-CoV-2 antibodies and associated factors in healthcare workers: A systematic review and meta-analysis. J. Hosp. Infect. 2021;108:120–134. doi: 10.1016/j.jhin.2020.11.008.
-Coppeta L, Ferrari C, Somma G, Mazza A, D'Ancona U, Marcuccilli F, Grelli S, Aurilio MT, Pietroiusti A, Magrini A, Rizza S. Reduced Titers of Circulating Anti-SARS-CoV-2 Antibodies and Risk of COVID-19 Infection in Healthcare Workers during the Nine Months after Immunization with the BNT162b2 mRNA Vaccine. Vaccines (Basel). 2022 Jan 18;10(2):141. doi: 10.3390/vaccines10020141.
REPLY: we really appreciated Reviewer’s suggestions. We have read with interest these articles which helped to further improve our discussion. We added the references in the bibliography in page 3, line 167 and 206.
Reviewer 2 Report
I considered the manuscript entitled “SARS-CoV-2 vaccination in solid-organ transplant recipients” by Maddalena Peghin, et al that is intended to be published in Vaccines journal.
Overall, the review is fine. Patients with SOT show characteristic and differential variations compared to the general population. It is interesting to update these data although their validity is short in the special field of Covid19
Some mild-major concerns must be addressed:
Reword please, it is a little confusing: Despite growing knowledge of the disease, the use of antiviral treatments and the 31 continuous use of non-pharmacological interventions the role of vaccination remains crucial, particularly for at‐risk populations. The rapid development of vaccines against the SARS-CoV-2, in association with other improvements have led transplant programs to return to pre-pandemic levels and to reduction of COVID-19 morbidity and mortality in SOT population [3, 4]. SARS-CoV-2 vaccines have proven to be safe in SOT recipients but with decreased immunogenicity and efficacy compared with the general population. However, the ongoing emergence of novel variants of concern (VOCs) remains a challenge, with reduced protection against infection, though SARS-CoV-2 vaccines seem to continue preventing severe illness resulting in hospitalization, intensive care admission, and death [5].
Is this the authors personal point of view? As an alternative, if 80 mRNA vaccines are not available or suitable because of contraindications, adenoviral vec-81 tor vaccines and inactivated whole virus vaccines should be recommended rather than 82 renunciation of COVID-19 vaccination
Expand this item, titles considered….: Althogh the neutralizing antibody level has more recently been established as a reliable predictor of protection against symptomatic COVID-101 19, the measures used in many studies varied
Reword: Is even necessary there? Of note that in studies including patients with a previous history of COV19 infections, seroconversion rates were sensibly higher even among unvaccinated patients, from 147 85% to 100%
There is a general conscious on the negative effect of immunosuppressive drugs in SOT and autoimmunity diseases on the response to vaccination. Authors should comment general mechanisms as the case of Rituximab is obvious as the worst drug because of direct CD20 action, but the other… A new concept is the importance of Belatacept. Is there a special mechanism?
Improve, too succinct: The best approach to SARS‐CoV-2 vaccination before and after transplant is not well known. All this chapter refers to authors personal point of view or come from specific Guidelines??
Line 251: patients on dialysis are systematically vaccinated independently of the waitlisting status. This sentence is not completely correct
Heterologous vaccination?
Author Response
REPLY: Dear reviewer, we thank you for your comments. We considered each point suggested, reported below and marked them within the text.
I considered the manuscript entitled “SARS-CoV-2 vaccination in solid-organ transplant recipients” by Maddalena Peghin, et al that is intended to be published in Vaccines journal. Overall, the review is fine. Patients with SOT show characteristic and differential variations compared to the general population. It is interesting to update these data although their validity is short in the special field of Covid19
Some mild-major concerns must be addressed:
Reword please, it is a little confusing: Despite growing knowledge of the disease, the use of antiviral treatments and the 31 continuous use of non-pharmacological interventions the role of vaccination remains crucial, particularly for at‐risk populations. The rapid development of vaccines against the SARS-CoV-2, in association with other improvements have led transplant programs to return to pre-pandemic levels and to reduction of COVID-19 morbidity and mortality in SOT population [3, 4]. SARS-CoV-2 vaccines have proven to be safe in SOT recipients but with decreased immunogenicity and efficacy compared with the general population. However, the ongoing emergence of novel variants of concern (VOCs) remains a challenge, with reduced protection against infection, though SARS-CoV-2 vaccines seem to continue preventing severe illness resulting in hospitalization, intensive care admission, and death [5].
REPLY: We thank the reviewer for his suggestion. In keeping with reviewer suggestion, we rephrased the introduction to clarify the aim of our review.
Is this the authors personal point of view? As an alternative, if 80 mRNA vaccines are not available or suitable because of contraindications, adenoviral vec-81 tor vaccines and inactivated whole virus vaccines should be recommended rather than 82 renunciation of COVID-19 vaccination
REPLY: The Reviewer is right to point out this issue. As explained in the paragraph 5 “Immunogenicity, clinical effectiveness and safety of SARS-CoV-2 vaccination in SOT” adenoviral vector vaccines and inactivated whole virus vaccines have showed to be clinical effective although fewer evidence is available. Moreover mRNA vaccines were approved before other vaccine types and existing studies focused on COVID-19 vaccination in SOT are mostly based on the use of mRNA vaccines. As such, we believe that it is reasonable to recommend their use whenever mRNA vaccines are contraindicated.
Expand this item, titles considered….: Although the neutralizing antibody level has more recently been established as a reliable predictor of protection against symptomatic COVID-101 19, the measures used in many studies varied
REPLY: We agree with the referee that this is an issue that requires further comments. There is still not a standardized agreement on the diagnostic methods and on the threshold of clinical efficacy of neutralizing antibodies. We have addressed this issue at page 3, lines 172-173
Reword: Is even necessary there? Of note that in studies including patients with a previous history of COV19 infections, seroconversion rates were sensibly higher even among unvaccinated patients, from 147 85% to 100%
REPLY: The Reviewer is right to point out this issue. The word “even” is used to explain that the study included patients with a history of COVID-19 both vaccinated and unvaccinated and that the rate of seroconversion after vaccination coupled with natural infection was higher than after only vaccination in both groups.
There is a general conscious on the negative effect of immunosuppressive drugs in SOT and autoimmunity diseases on the response to vaccination. Authors should comment general mechanisms as the case of Rituximab is obvious as the worst drug because of direct CD20 action, but the other… A new concept is the importance of Belatacept. Is there a special mechanism?
REPLY: We agree that this is an interesting topic. As explained by Chavarot et al (Weak antibody response to three doses of mRNA vaccine in kidney transplant recipients treated with belatacept. Am J Transplant. 2021 Dec;21(12):4043-4051. doi: 10.1111/ajt.16814.) “belatacept had been shown to play a direct and active role at several steps of the humoral response by reducing plasmablast differentiation, Ig production, and the ex- pression of the major transcription factor involved in plasma cell function, Blimp-1, in a T cell–independent manner. Moreover, belatacept modulates B cell-Tfh crosstalk, leading to an impaired germinal center formation and an improper antibody response in KTRs treated with belatacept. Interestingly, a recent study reporting immune responses to mRNA vaccines, showed that most spike-specific Th cells expressed the coactivating molecule CD28; and could therefore be theoretically be inhibited by belatacept.”
Improve, too succinct: The best approach to SARS‐CoV-2 vaccination before and after transplant is not well known. All this chapter refers to authors personal point of view or come from specific Guidelines?
REPLY: The Reviewer is right to point out this issue. The experience with previous vaccinations and the understanding of the immune response to vaccines in SOT, makes it reasonable to recommend vaccination before the transplant when feasible, to reach a higher immune response before the post-transplant immunosuppression. Of note that patients with chronic liver disease and end-stage renal disease have shown a poor humoral and cellular immune response to SARS-CoV-2 vaccination compared with the general population. However, waitlisted patients who have been vaccinated exhibit a more robust response to vaccines, compared with the poor response in naive recipients vaccinated after transplantation
Line 251: patients on dialysis are systematically vaccinated independently of the waitlisting status. This sentence is not completely correct.
REPLY: We agree that this is a topic that requires explanation. End-stage renal disease patients should be vaccinated against SARS‐CoV-2, independently if they are waitlisted or not, due to higher risk for significant morbidity and mortality compared with the general population
Heterologous vaccination?
REPLY: We agree with the referee that this is a topic that requires explanation. We added the meaning of heterologous vaccination in page 6, line 252-253
Round 2
Reviewer 2 Report
None